# [Re] Fair Selective Classification Via Sufficiency

## Reproducibility Summary

**Scope of Reproducibility**

Bu, Lee et al. (2021) introduced a method for enforcing fairness in selective classification, deriving a novel upper bound for the conditional mutual information from the sufficiency criterion. We attempt to verify the second claim that: "[this novel upper bound] can be used as a regularizer to enforce the sufficiency criteria, [and] then **show that it works to mitigate the disparities on real-world datasets**."[4]

**Methodology**

To verify the author's claim, we implemented the model and regularizer described in the original paper. We wrote the code from scratch, since there was no code available. We train both a baseline and regularized model on three of the four datasets used by the authors: Adult, CelebA, and CheXpert. The Civil Comments dataset which the authors also used was computationally too expensive. We trained using the Adam optimizer and a constant learning rate. The training of the regularized models for Adult, Chexpert and CelebA takes under a minute, under 2 hours and under 4 hours respectively on a single NVIDIA Titan RTX.

**Results**

We found that we could not reproduce the original paper's results. While the area between precision curves decreases somewhat for the CelebA and CheXpert datasets, it increases for the Adult experiment. Also, the analysis of our margin distributions between the baseline and regularized models do not seem to indicate an increase in overlap between groups. Due to these results, we cannot conclude the effectiveness of the regularizer in reducing the disparity between two groups demonstrated by the authors of *Fair Selective Classification via Sufficiency*, using our implementation.

**What was easy**

Implementing the regularizer and group specific models in PyTorch was relatively straight forward, using the well documented loss functions and Algorithm 1 from the original paper [4].

**What was difficult**

We found that the results for all datasets were sensitive to preprocessing and hyperparameter tuning. Since the authors specified very little in this regard, experimenting with the dataset specific preprocessing steps, and the hyperparameter tuning for three datasets took us a considerable amount of time.

**Communication with original authors**

The authors of the original paper were emailed with multiple questions about preprocessing, training and the baseline. Unfortunately, because they also had an important deadline, they responded three days before the deadline giving us little time to make changes.

# 1 Introduction

Machine learning algorithms are being used to solve more and more diverse problems, and are fulfilling tasks in increasingly difficult situations. One way to improve the performance of classification models is to use selective classification [4]. This means that models are allowed to abstain when their prediction confidence is low. However, abstaining does come at the cost of coverage (the ratio of samples for which a decision is made). Previous work has shown that classifying selectively does not always affect all distinguishable groups within the data evenly, for instance in the CelebA[1] and CivilComments[2] datasets [3]. Implementing a model that is able to classify selectively in a fair way, not discounting certain groups within the data therefore is an active challenge for the artificial intelligence research community. The authors of the *Fair Selective Classification Via Sufficiency* paper [3] propose a method to improve fairness in selective classification accuracy between groups by using the sufficiency criterion.

The contribution of the authors of this paper is twofold. Firstly, they prove a novel upper bound for conditional mutual information. Secondly, they use this result to introduce a regularization technique that forces a model to be more fair to all protected groups when classifying selectively. These protected groups can be selected based on sensitive attributes (e.g. race, gender). They report improved overall group specific performance relative to a baseline method where they only optimize the cross-entropy loss. Furthermore, they improve relative to the group DRO method which has been shown to mitigate the disparity in recall rates between groups in selective classification [5]. In this study we verify the second claim that "[... their regularizer] works to mitigate the disparities on real-world datasets" by building a sufficiency regularized classifier that is more fair to underrepresented groups in selective classification.

# 2 Scope of reproducibility

In this paper we aim to reproduce the second claim from the original paper, which states: sufficiency can be used to train fairer selective classifiers which ensure that precision always increases for all groups as coverage is decreased. The authors support their claim by evaluating on the positive predictive parity, also called precision, by looking at the area under the curve for the accuracy for two groups within the Adult[4], CelebA, CheXpert[5] and CivilComments datasets. Since we found the CivilComments dataset and corresponding model to be too computationally expensive, we aim to reproduce the results on the first three datasets. The original authors did not publish any of their code. The scope of this reproducibility report is thus to write all necessary code and train and evaluate both the baselines and the regularized models for the Adult, CelebA and CheXpert datasets.

# 3 Methodology

This section discusses the methodology and experimental setup used to reproduce the paper *Fair Selective Classification via Sufficiency*. Firstly, the model is discussed, after which we go over the dataset specifics, the evaluation metrics, and the computational requirements.

## 3.1 Model description

The architecture comprises of three distinct components: the featurizer, classifier and regularizer. Using these components, the modelling objective can be described as finding model parameters $\theta_T$ and $\theta_\Phi$ for the classifier and featurizer such that the following equation is satisfied:

$$\min_{\theta_T, \theta_\Phi} \frac{1}{n} \sum_{i=1}^{n} \left( L\left(T\left(\Phi\left(x_i\right)\right), y_i\right) + R(\Phi(x_i), y_i, \theta_{d_i}, \theta_{\widetilde{d}_i}) \right) \tag{1}$$

Where $\Phi(x)$, $T(\Phi(x))$ and $R(\Phi(x), y, \theta_d, \theta_{\widetilde{d}})$ represent the featurizer, classifier and regularizer respectively. These individual components will be discussed in the following sections. An overview can be found in Figure 1. The loss function $L$ is not specified in the original paper, therefore we assume the cross-entropy loss is used, due to its popularity

---

[1]https://mmlab.ie.cuhk.edu.hk/projects/CelebA.html

[2]https://www.kaggle.com/c/jigsaw-unintended-bias-in-toxicity-classification/

[3]https://proceedings.mlr.press/v139/lee21b.html

[4]https://archive.ics.uci.edu/ml/datasets/Adult

[5]https://stanfordmlgroup.github.io/competitions/chexpert/

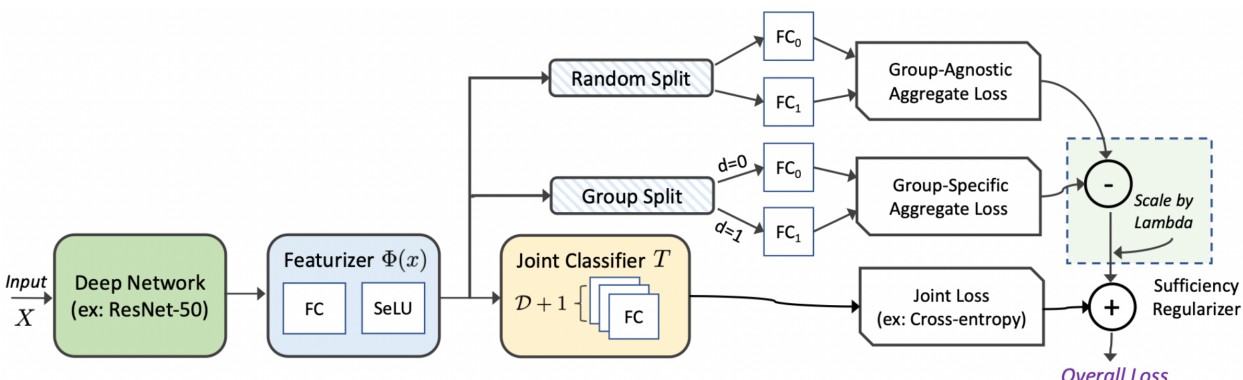

Figure 1: The model architecture from the original paper.

This entire Fair Selective Classifier ensures an improvement in precision for all groups as coverage decreases by applying the sufficiency criteria to the learned features. This sufficiency is enforced by regularizing the model through a novel upper bound of the conditional mutual information (CMI). In this section we describe the classifier architecture including the CMI regularization.

### 3.1.1 Featurizer

The featurizer $\Phi(x)$ for each dataset is trained to be predictive about Y while allowing for the classification to be calibrated by group based on the sensitive features. Under that constraint for all groups $d \in \mathcal{D}$ we have that classification performance for specific groups is never sacrificed to increase overall performance. This part of the architecture is dataset specific.

### 3.1.2 Classifier

The structure of the *joint classifier* $T(\Phi(x))$ is vaguely specified in the proposed architecture, specifying a number $D + 1$ and displaying a parallel stack of fully connected parallel layers in Figure 2 of the original paper. We chose to implement it as a single linear layer taking as input the extracted features from the featurizer. The output size is dependent on the classification task, with a single node for binary classification and $n$ nodes for multi-categorical classification. The output of this part of the model is used for inference and therefore also final evaluation.

### 3.1.3 Regularizer

In order to impose the *sufficiency condition*, a regularization term $R(\Phi(x), y, \theta_d, \theta_{\widetilde{d}})$ is added to the joint classification loss. This regularization loss ( Equation 2) is calculated according to the following equation.

$$R(\mathbf{x}, y_i, \theta_{d_i}, \theta_{\widetilde{d}_i}) = \lambda \log q\left(y_i \mid \Phi\left(x_i\right); \theta_{d_i}\right) - \lambda \log q(y_i \mid \Phi\left(x_i\right); \theta_{\widetilde{d}_i}) \tag{2}$$

The architecture achieves this in the following way. Firstly, the sample features are split in two ways: based on their true sensitive attribute values $d$ (the group split) and based on a randomized sensitive attribute $\widetilde{d}$ (the random split), where $\widetilde{d}$ is sampled randomly from empirical distribution $\hat{P}_d$. Both these splits are subsequently activated by the same attribute-specific linear layers $q\left(y \mid \Phi(x); \theta_d\right)$ to perform group-specific and group-agnostic classification. After classification, the loss term $L_d$ is calculated for each split using the cross-entropy loss, scaled by $\lambda$, and subtracted to capture the difference (Equation 2).

### 3.2 Datasets

This section goes over each dataset, and gives a description of the data, the selected sensitive attribute, and the featurization. For the experiments we used the same datasets as the authors. To handle the the different types of datasets, four different featurization architectures are used. For a more detailed description of the datasets see Appendix A.

The Adult dataset[6] records census data. Each data point has both continuous attributes and categorical attributes. The goal for this dataset is to predict whether an individual makes over 50k per year. The selected sensitive attribute for this dataset is sex. To preprocess the data we normalized the continuous attributes to have zero mean and unit variance. We introduced a bias in the data the same way as in the original paper by removing all but the first 50 rows for which the protected group $D = 0$ and the target $Y = 1$ (what this means is unclear from the paper; we used "Female" and ">50K" respectively). The Adult Dataset comprises of tabular data, and therefore the featurization component is a single linear layer. The layer has 80 output nodes and is followed by the SeLU activation function.

The CelebA dataset[7] contains RGB images of celebrities. Each image is provided with annotations about the appearance of the celebrities. The task at hand is to predict whether a celebrity has blond hair and the selected sensitive attribute is sex. We followed the original paper and resized the images to 224 by 224. The ResNet-50 model has it's weights pre-trained on ImageNet and the classification layer is dropped to output the 2048 features of the penultimate layer.

The CheXpert dataset[8] consists of chest x-ray images and corresponding attributes annotated by experts. The task for this experiment is to predict the presence of pleural effusion (a lung disease), and the sensitive attribute is the presence of a support device. To preprocess CheXpert we removed all data points where either the sensitive attribute or the target was labeled uncertain. Again following the original paper, the x-ray images were all resized to 224 by 224 pixels and stacked to simulate the red, green and blue color channels that Densnet121 expects. For the CheXpert image dataset a DenseNet-121 model (pre-trained on image net) is used as featurizer. The classification layer is dropped and an AvgPool2d layer is added to output the penultimate layer's feature vector of size 1024, in accordance with the DenseNet121 architecture [2].

The Civil comments dataset[9] contains comments on news articles. For this dataset the task is to predict whether a comment was toxic or not. The selected sensitive attribute was the commenter being Christian. To preprocess the data, all comments with unknown religious background were removed. The comments were tokenized using the BERT tokenizer and the tokenized comments were padded to be able to fit the dimensions of the BERT model as it takes tensors with a size of 512 tokens. The final classification layer of the BERT-model is replaced with an extra linear layer to output a feature vector of size 80. The specific version of BERT was not specified, it can be safe to assume that it was the uncased BERT-Base. The model can be downloaded from the PyTorch-Transformers library. We chose a model pre-trained for sequence classification.

During experimentation we concluded that the computational requirements were too high for our time constrains and setup. We therefore decided to train models for the Adult, CelebA and CheXpert dataset, but we provide code for CivilComments as well.

### 3.3 Hyperparameters

The original paper specifies $\lambda = 0.7$ and the dataset specific number of epochs. The learning rates and optimizer were not specified. Therefore, we decided to grid search the learning rate(s). They use three different learning rates in their notation: $\eta_d, \eta_f, \eta$ for the group specific models, the featurizer and the joint classifier respectively. However, trying multiple different combinations of these learning rates would require a lot of computational resources, so we decided to only perform a grid search over a general learning rate $\boldsymbol{\eta}$, which sets the same value for $\eta_d, \eta_f$ and $\eta$.

In the process of finetuning the model we found that depending on the dataset learning rates between 0.001 and 0.0001 led to stable and generalizing models. Shortly before the submission deadline for the Machine Learning Reproducibility Challenge 2021 we were informed by the authors that they used the Adam optimizer, and used a learning rate of $0.001$. In Table 1 we specify the entire set of hyperparameters for every per dataset.

### 3.4 Experimental setup

In this section we go over the experimental setup. We first explain the training and the evaluation process. After that, we also go over the hardware and software used for this study.

---

[6]https://archive.ics.uci.edu/ml/datasets/Adult
[7]https://mmlab.ie.cuhk.edu.hk/projects/CelebA.html
[8]https://stanfordmlgroup.github.io/competitions/chexpert/
[9]https://www.kaggle.com/c/jigsaw-unintended-bias-in-toxicity-classification/

| Parameters | Adult | CelebA | CheXpert |
|---|---|---|---|
| $\lambda$ | {0, 0.7} | {0, 0.7} | {0, 0.7} |
| $\eta$ | 0.001 | 0.001 | 0.001 |
| nr. epochs | 20 | 10 | 10 |
| batch size | 32 | 128 | 64 |

Table 1: The set of hyperparameters for every dataset.

### 3.4.1 Model Training

During model training, we alternate between two backpropagation steps following the original paper. Firstly we fit the group-specific models for each batch and secondly we update the feature extractor and joint classifier.

### 3.4.2 Evaluation metrics

This section goes over the evaluation metrics of the original paper by first discussing the confidence score and the margin. These concepts are then used to derive the final evaluation metrics: the area under the accuracy curve and the area between precision curves. For more details, see Section 2 of the original paper.

The classifier has the possibility to abstain from the decision based on a confidence score $\kappa(x)$ and a threshold $\tau$. The used confidence score is defined as the monotonic mapping of the softmax response $s(x)$

$$\kappa(x) = \frac{1}{2} \log \left( \frac{s(x)}{1 - s(x)} \right) \tag{3}$$

which maps [0.5, 1] to [0, $\infty$] to provide a high resolution on the values close to 1. Since we can map a softmax response to the interval [0.5, 1] for both targets ($s(x) = s(x)$ for $s(x) >= 0.5$, and $s(x) = 1 - s(x)$ for $s(x) < 0.5$), it is possible to use this function.

The confidence score is used to define the margin $M$, such that is defined as $M(x) = \kappa(x)$ if $\hat{y}(x) = y$ and as $M(x) = -\kappa(x)$ otherwise. If we then use $\tau$ as our threshold for abstaining, the selective classifier makes correct predictions when $M(x) \geq \tau$ and incorrect predictions when $M(x) \leq -\tau$.

The model is evaluated with different values of $\tau$. The selective accuracy is computed for the different coverages, caused by the different values of $\tau$. The selective precision is computed similarly, conditioning on $\hat{Y} = 1$. To measure the effectiveness of the selective classifier at different coverage levels, the area under this curve is computed. The difference in precision across groups sometimes reveals some disparities that are not revealed by only considering the difference in accuracy. Therefore the precision-coverage curves are also plotted per group. The difference between these curves is computed to encapsulate the difference across different coverages.

### 3.4.3 Computational requirements & Code

All training is done on GPU nodes of a cluster which contains multiple kinds of Nvidia GPU's. For our training we used the Titan RTX nodes. These GPU's have 24GB of GDDR6 memory. The training of the regularized model took under 2 minutes for Adult, under 2 hours for Chexpert and under 5 hours on the CelebA dataset. To be able to store all the datasets around 20GB of storage is necessary.

All code used for the data preprocessing, model training and model evaluation can be found in our Github repository[10].

## 4 Results

### 4.1 Area under the Curve & Area between Precision Curves

Table 2 lists the area under the accuracy curve (*auc*) and the area between precision curves (*abc*) for the baseline and regularized model for every dataset. The change in *auc* shows that the introduction of the regularization does not harm (or even improves) the overall performance of the models for all datasets. This is in line with the results of the original paper.

---

[10]https://anonymous.4open.science/r/FSCS-4F57/README.md

| Dataset | Method | Area under accuracy curve | Area between precision curves |
|---|---|---|---|
| Adult | Baseline | $0.93 \pm 0.0002$ | $0.056 \pm 0.0008$ |
|  | Regularized | $0.93 \pm 0.006$ | $0.065 \pm 0.009$ |
| CelebA | Baseline | $0.93 \pm 0.040$ | $0.010 \pm 0.0002$ |
|  | Regularized | $0.99 \pm 0.0001$ | $0.006 \pm 0.0001$ |
| CheXpert | Baseline | $0.83 \pm 0.013$ | $0.075 \pm 0.036$ |
|  | Regularized | $0.84 \pm 0.02$ | $0.070 \pm 0.0009$ |

Table 2: Area under accuracy curve results for all datasets

Our results for the *abc*, however, diverge. The original paper reports a decrease for the *abc* by a factor of 10, 9 and 2 going from the baseline to the regularized model, for Adult, CelebA and CheXpert respectively. The *abc* of our regularized model for Adult is higher than its baseline, and for CelebA and CheXpert the improvement is a factor of 2 and 1.1 over the baseline. Another point of interest is the high standard deviation for the CheXpert baseline which makes the improvement of the regularized model questionable.

## 4.2 Margins and Precision-Coverage Plots

The dataset specific margins of both the baseline and the sufficiency regularized model can be found in Figure 2 and the precision-coverage plots can be found in Figure 3. Our plots show different characteristics compared to the original paper, and the difference between the unregularized and regularized models is small for all datasets.

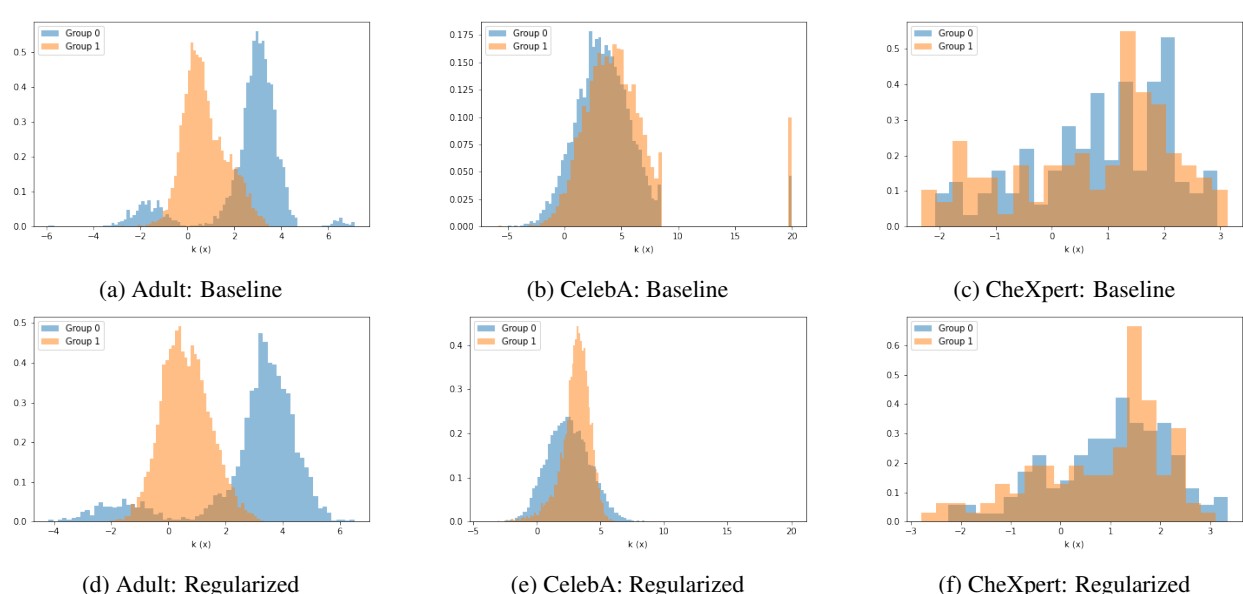

(a) Adult: Baseline    (b) CelebA: Baseline    (c) CheXpert: Baseline

(d) Adult: Regularized    (e) CelebA: Regularized    (f) CheXpert: Regularized

Figure 2: Margin distributions for the datasets for the baseline and regularized methods. For CelebA, all confidence scores were capped to 20.

## 5   Discussion

Table 2 shows the method comparison between the baseline and regularized models for each dataset. While *abc* values decrease somewhat for the CelebA and CheXpert datasets, it increases for the Adult experiment. The margins and precision-coverage plots of the original paper show a clear improvement for the worst case group going from the baseline to the regularized version. Their regularized models show more overlap between the distributions of the two groups, and the precision-coverage curves are closer to each other for the regularized models than for the baseline. The

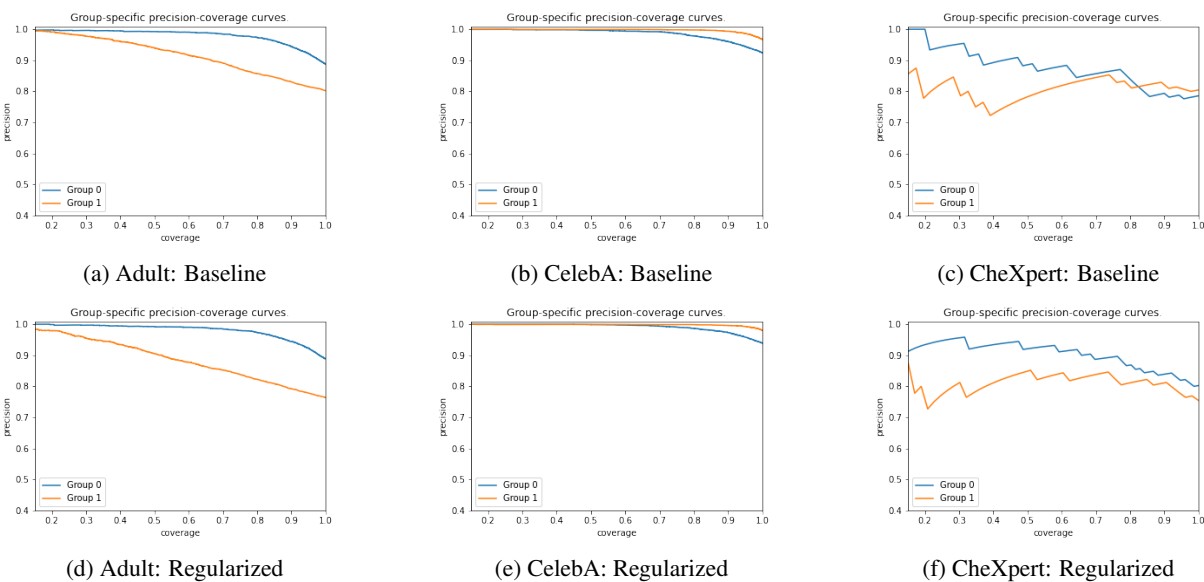

Figure 3: Group-specific precision-coverages curves for the baseline and regularized methods.

analysis of our margin distributions between the baseline and regularized models (Figure 2) do not seem to indicate an increase in overlap between groups. This means that these margin and precision-coverage comparisons for our experiments do not give a conclusive result on the effect of regularizing the classifier.

Due to these contrary and/or inconclusive results on the impact on the regularization on selective classification across datasets, we cannot conclude its effectiveness in reducing the disparity between two groups demonstrated by the authors of *Fair Selective Classification via Sufficiency* using our implementation.

## 5.1 What was easy

Implementing the regularizer and group specific models in PyTorch was relatively straight forward, using the well documented loss functions and Algorithm 1 from the original paper. It was also relatively easy to configure the featurizers for each dataset as they were clearly described as well. Implementing the architecture was straight forward using the clear figure from the original paper.

## 5.2 What was difficult

We found that the results for all datasets were sensitive to preprocessing and hyperparameter tuning. Since the authors specified very little in this regard, experimenting with the dataset specific preprocessing steps, and the hyperparameter tuning for three datasets took us a considerable amount of time.

## 5.3 Communication with original authors

We sent an email to the original authors. We asked them about preprocessing the CheXpert and CelebA dataset because we didn't know exactly how they preprocessed the images in this dataset and how they used some of the attributes. We also wondered how Figure 2 in the original paper should be interpreted.

Furthermore we had questions about training the models. We did not know what learning rates and optimizers they used for training. Knowing this would have saved us from the time-consuming task of running a grid search. Shortly before our submission deadline, we received an email from the authors with answers to our questions. This gave us some insights into which hyperparameters and optimizer they used. This was just in time to rerun all models for multiple seeds.

## Acknowledgement

We would like to thank [name to be added] for his guidance during the replication process.

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

# A Dataset Analysis

## A.1 Descriptive analysis datasets

### A.1.1 Civil Dataset

The Civil Comments dataset consists of comments on news article's which were collected on the Civil Comments platform. This is a platform in which people are able to post a comment after they verify two other comments. This makes the users self moderate the comments on the platform. All comments are annotated by multiple users and the toxicity score is an average of binary toxicity classifications. In figure 4 we plotted the amount of comments for each length. As you can see there is a clear spike of comments at a length of 1000 words. This can be explained due to a maximum amount of words per comment. Because the BERT model takes 512 words at maximum we truncated every comment which was longer than this.

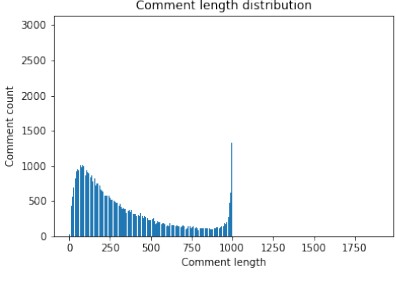

(a) Distribution of comment length

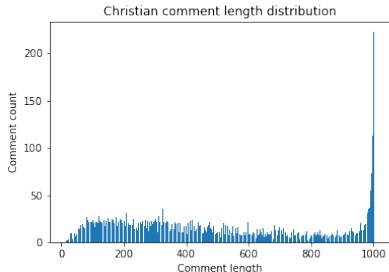

(b) Distribution of Christian comment length

Figure 4: Two comment length distributions of the civil comments dataset

The Civil comments dataset was available through a challenge on the Kaggle dataset platform [11]. It contains 1971916 comments from the Civil comments platform. This was a platform created to make comments more civilized by letting users that want to comment first rate other people's comments. The dataset was split into a training set and a test set, with 90 percent of the data being the training set and 10 percent the test set. Each comment row contained information about other peoples like reactions, multiple forms of toxicity and information about the person who commented. For this dataset the task is to predict whether a comment was toxic or not. The selected sensitive attribute was the commenter being Christian which is reported as a binary Christian or not Christian (1 or 0, respectively).

The following steps were taken to preprocess the data. In 78 percent of the comments it was not known whether the commenter was Christian or not. Therefore these comments were removed from the dataset and it left us with 370646 comments in the train set and 5811 comments in the test set. The comments also need to be tokenized to be able to use it for the BERT model and we used the BERT tokenizer to do this. The tokenized comments also needed to be padded to be able to fit the dimensions of the BERT model as it takes tensors with a size of 512 tokens.

### A.1.2 CelebA Dataset

The CelebA dataset was obtained through the Large-scale CelebFaces Attributes Dataset website [12] but it was stored on Google Drive. The dataset contains 202599 RGB images of 10177 celebrities. The dataset was split into a training set of 162770 images, a validation set of 19868 images and a test set of 19961 images. There were three types of annotations available: landmark annotations, attributes annotations and identities annotations. We only used the attribute annotations, these contain annotations about their appearance. The task at hand is to predict whether a celebrity has blond hair. The selected sensitive attribute (by the original paper) is sex just like in the Adult Dataset. It is reported as a binary for the attribute "Male".

---

[11]https://www.kaggle.com/c/jigsaw-unintended-bias-in-toxicity-classification/
[12]https://mmlab.ie.cuhk.edu.hk/projects/CelebA.html

The only necessary preprocessing step was to resize the images. All images in the dataset have the dimension 178 x 218 x 3. But ResNet50 takes dimension 224 x 224 x 3 as input. Therefor each image had to be resized to this dimension.

To get some insights into the data we decided to calculate the average images for some specific subgroups as you can see in figure 5. In subplot 5b you can clearly see a female average for all celebrities with blond faces. This was confirmed when we calculated the percentage of blond celebrities and the percentage of blond male celebrities. From the complete dataset 14 percent was blond and of this subgroup 5 percent was male. This clearly indicates a minority group in the dataset.

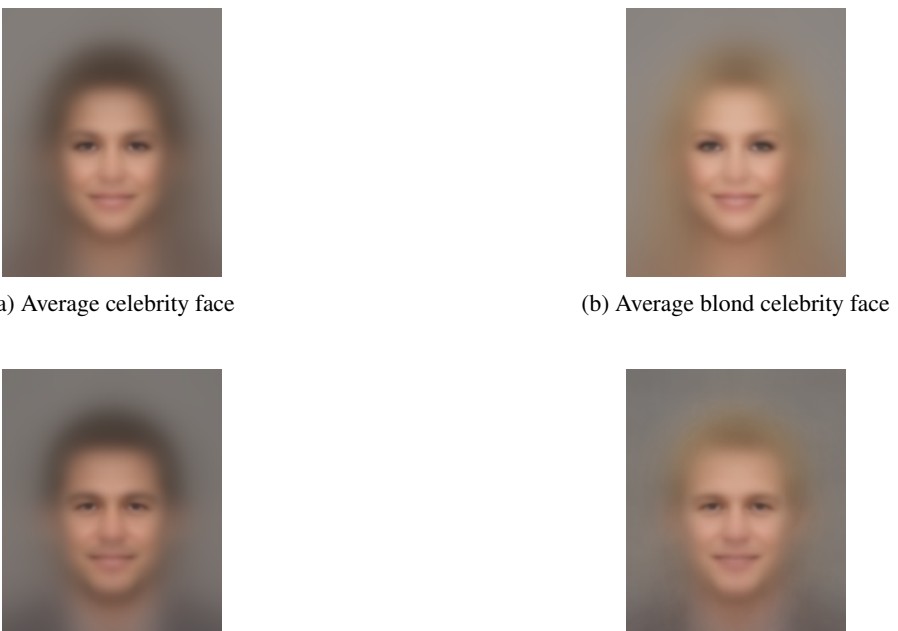

(a) Average celebrity face

(b) Average blond celebrity face

(c) Average male celebrity face

(d) Average blond male celebrity face

Figure 5: Average faces of different subgroups of the dataset

### A.1.3 Adult Dataset

The Adult dataset was obtained through the AI Fairness 360 toolkit [1]. The dataset records census data of 45222 individuals (after removing incomplete rows), split in a training set of 29100 and a test set of 15060 data points. Each data point has both continuous attributes such as age, capital gain and capital loss, and categorical attributes such as sex, marital status and native country. The task corresponding to this dataset is to predict whether each individual earns more than 50K per year. The selected sensitive attribute (by the original paper) for this dataset is sex which is reported as a binary: "Male" and "Female".

We took the following steps to preprocess the data. Firstly, we normalized the continues attributes to have zero mean and unit variance. Secondly, we introduced a bias in the data the same way as in the original paper by removing all but the first 50 rows for which the protected group $D = 0$ (our experiments showed that this was probably "Female"), and the target is ">50k". This removes 1062 samples from the training set.

### A.1.4 Chexpert Dataset

We obtained the small CheXpert dataset from the website of the Stanford ML Group [13]. This dataset consists of chest x-ray images and corresponding attributes annotated by experts. These attributes indicate the presence of diseases (pathology), which are all possible classification targets, and the presence of a support device. The task for this experiment is to predict the presence of pleural effusion, and the chosen sensitive attribute is the presence of a support device.

---

[13]https://stanfordmlgroup.github.io/competitions/chexpert/

The preprocessing of CheXpert was relatively straight forward. We removed all data points where either the sensitive attribute or the target was labeled uncertain. The x-ray images were all resized to 224 by 224 pixels and stacked to simulate the red, green and blue color channels that Densnet121 expects.

