# OpenReview forum: "[Re] Fair Selective Classification Via Sufficiency"
_ML_Reproducibility_Challenge/2021/Fall — Reject_

### Official Review · Reviewer_zP1c · 2022-03-07
**Thorough replication, limited discussion.**

**Rating:** 6
**Confidence:** 4

**Review:**

The authors do a good job of detailing the setup of the paper that is being reproduced, and their reproduction efforts. However, I found the actual results and discussion to be very limited. Given the (very nice) thoroughness that the reproduction has in describing the setup of the paper and the process for reproduction, I was surprised to see that there is a relatively sparse discussion of the results. With that being said, I do think the authors do a nice job of discussing the sensitivity of the method to hyperparameter tuning and preprocessing, and I think that the overall quality of presentation is quite high.

---

### Official Review · Reviewer_shMQ · 2022-03-20
**Lack important details and explanations, report not reproducible at the moment**

**Rating:** 4
**Confidence:** 4

**Review:**

Scope and objectives of this reproducibility challenge report are clear, but more details on the methodological set up are needed. The authors should describe all the settings that were tried, so that results from the report are reproducible as well.

Some examples of missing methodological descriptions are: how many random inits? how was the train/test split performed, are you using cross validation? What are the hyperparameters that were used? How many epochs? Etc are missing.

Re clarity: it would be nice if the report was more self-consistent (define in lay terms what are each hyperparameter that you are tuning for example). Also results are said to have "different characteristics" from those reported in the original paper, but no explanation is provided, please clarify.

Other comments: Some claims need justification: e.g., "learning rates between so and so led to stable and generalizing models (as measured by what)". Table 1 shows set of hyperparameters: are these the best selected ones? It would be useful to have a table to show the grid search used for each param. Important details about the experimental setup are missing, Table 2: how is the standard deviation computed? Is this trained on different random splits? Are you usinf cross-validation? Figure 2 lacks explanation, it just says distributions are different. Re hyperparameters, remind the reader of the report what is lambda, and others. Make Equations self contained by introducing all sub-indices. What is \tilde{d_i} for example in Equation (1)? Correct punctuation and grammatical errors.

Finally, in the anonymous github, it would be helpful if the authors include one file with the exact final experiments, and how to reproduce the results in their report, honoring the theme of the reproducibility challenge.

---

### Meta-Review · Area_Chair_5dEy · 2022-04-09

**Recommendation:** Reject
**Confidence:** 4

**Metareview:**

Reviewers remarked that the reproduction didn't contain enough detail (for example the list of hyperparameters). They also asked for more discussion.

---

### Decision · Program_Chairs · 2022-04-09

Reject